# Categorising Subjective Perceptions of Middle-Aged Breast Cancer Patients Using Q Methodology

**DOI:** 10.3390/healthcare12181873

**Published:** 2024-09-18

**Authors:** Min-Jeung Shim, Song-Yi Lee, Oh-Sun Ha

**Affiliations:** 1Department of Counselling and Coaching, Dongguk University-Seoul, Seoul 04620, Republic of Korea; 2021121157@dongguk.edu; 2Academic Affairs Team, Dongguk University-Seoul, Seoul 04620, Republic of Korea

**Keywords:** middle-aged, breast cancer, patients, women, subjective

## Abstract

Background/Objectives: This study explores the characteristics of different perception types in middle-aged female breast cancer patients and proposes psychological counselling interventions tailored to each type. Methods: The study used the Q methodology, starting with the construction of 40 Q samples and 39 P samples. Results: We categorised middle-aged female patients’ subjective perceptions of battling breast cancer into five types along a spectrum: Type 1, ‘Embracing a New Life’; Type 2, ‘Finally Focusing on Myself’; Type 3, ‘Struggling Amidst Confusion’; Type 4, ‘Withdrawing in Despair’; and Type 5, ‘Pushed to the Edge of Fear’. Conclusions: This study revealed that the subjective experiences of middle-aged female breast cancer patients range on a spectrum from post-traumatic growth to post-traumatic stress disorder (PTSD). Based on these findings, this study discusses the characteristics and significance of each subjectivity type and suggests implications and directions for future research.

## 1. Introduction

Cancer, responsible for 22.4% of all deaths and the leading cause of mortality in South Korea, is a prominent intractable disease [1]. There is a notable upward trend in the incidence of female cancers, with women accounting for 48.2% of all cancer cases in 2021 [2]. Among these, breast cancer has the highest incidence rate and poses a significant threat to women worldwide [3,4]. Breast cancer patients typically undergo mastectomy or lumpectomy to treat the disease and prevent recurrence. Adjuvant therapies such as radiation, hormone therapy, chemotherapy, and targeted therapy often follow this procedure. These treatments cause physical suffering, including pain, nausea, hot flashes, and exacerbated menopausal symptoms [5,6]. In addition to physical pain, breast cancer patients also face significant psychological challenges, such as feelings of despair, anxiety about recurrence, depression, and anger [7,8,9,10,11,12]. Mastectomy, in particular, can be a major shock to patients, leading to feelings of loss and decreased self-confidence due to changes in body image and perceived loss of femininity [13,14]. These physical and psychological difficulties can affect family, social, and work life, leading to isolation and exacerbating psychological distress [15,16,17,18].

Breast cancer has various risk factors, including genetics, age of menarche and menopause, and dietary habits [2]. It frequently occurs in women aged 40–59, highlighting age as a major risk factor [19]. Unlike Western countries, where breast cancer often occurs post-menopause, many breast cancer patients in Korea are younger, in the 40–65 age range, i.e., middle-aged [3]. Middle age is a period characterised by significant biological, psychological, and cognitive changes, often leading to psychological instability and crises [20,21]. These age-specific challenges, coupled with the difficulties of battling breast canfcer, exacerbate psychological instability and suffering [22,23,24]. Consequently, many middle-aged female breast cancer patients experience considerable distress, with some studies identifying these reactions as post-traumatic stress disorder (PTSD) [9,25]. Research indicates that 44% of breast cancer patients exhibit PTSD symptoms shortly after diagnosis [26], and the prevalence of acute stress disorder (ASD) and PTSD among breast cancer patients is approximately 2.4% [25].

However, not all middle-aged female breast cancer patients experience extreme psychological instability; some undergo post-traumatic growth, exhibiting positive changes and a renewed appreciation for life [27,28,29]. Compared to the general female population, breast cancer patients report higher levels of post-traumatic growth, including improved relationships, new possibilities, personal strength, appreciation of life, and spiritual change [28,30]. As such, middle-aged female breast cancer patients may experience post-traumatic stress disorder due to severe psychological instability and pain, but they may also perceive the breast cancer battling process as a turning point in their lives, experiencing post-traumatic growth personally and in their outlook on life. We should understand these experiences on a spectrum, with extremes of post-traumatic stress disorder and post-traumatic growth. Therefore, for a comprehensive understanding of breast cancer patients, it is essential to consider their physical conditions from a medical standpoint and their psychological and emotional states [31,32,33,34].

Most current studies on breast cancer patients focus on therapeutic or nursing approaches to reduce physical suffering and manage side effects [7,13,14,15,35,36,37]. While addressing the physical aspects is crucial, it is equally important to recognise the diverse psychological experiences of these patients. Research based solely on medical interventions may fall short of providing a holistic understanding of breast cancer patients. Therefore, this study provides a comprehensive understanding of middle-aged female breast cancer patients by examining their psychological experiences.

This research employs the Q methodology to categorise and characterise the subjective experiences of these patients. It proposes tailored psychological counselling interventions for each identified experience type. The results of this study deepen our understanding of the experiences of middle-aged women battling breast cancer and inform strategies for psychological and counselling support. The results will facilitate discussions on effective psychological and counselling interventions tailored to these patients.

## 2. Research Method

The Q methodology is a methodological approach that reveals subjective perceptions by examining individuals’ subjective concepts, viewpoints, beliefs, and attitudes regarding specific situations or contextual factors [38,39]. Its goal is to group participants based on the overall coherence of their responses and extract factors [40,41]. Thus, it can measure subjective perceptions based on individual internal values [40,41,42]. This approach is suitable for the study as it allows for an objective and in-depth understanding of patients’ experiences. Figure 1 illustrates this study’s Q methodology procedure.

### 2.1. Q Population Construction and Q Sample Selection

The Q population encompasses a comprehensive range of thoughts on the subject, which we gathered through literature reviews and interviews [42]. This study developed the Q population through literature reviews, preliminary interviews, and surveys. Initially, for the literature review, we performed electronic searches using keywords such as breast cancer, post-traumatic growth, post-traumatic stress disorder, and treatment process of breast cancer patients via Google Scholar and RISS, which is a research information sharing service in Korea. Based on these keyword searches, we selected relevant academic journals and dissertations for analysis [7,13,15,43]. From these sources, we identified 30 statements.

Next, we developed a semi-structured questionnaire and conducted preliminary interviews with ten middle-aged female breast cancer patients. The participants included three individuals in their 40s, six in their 50s, and one in her 60s. By cancer stage, one participant was in stage 1, four in stage 2, four in stage 3, and one in stage 4, with treatment durations ranging from seven months to seven years. The semi-structured questionnaire used in the preliminary interviews included questions such as “What does your experience of fighting breast cancer mean to you?” and “In which area of your life have you experienced the most significant change?”.

Finally, based on the revised semi-structured questionnaire from the preliminary interviews, we conducted a subjective survey with 98 middle-aged female breast cancer patients aged 40 to 64 who were undergoing treatment. The participants included 52 individuals in their 40s, 39 in their 50s, and seven in their 60s. The distribution by cancer stage was as follows: 34 in stage 1, 38 in stage 2, 23 in stage 3, and three in stage 4.

Through the preliminary interviews and subjective surveys, we collected 1452 statements. Combined with the 30 statements extracted from the literature, we had 1482 statements. The researchers repeatedly read and categorised these statements. As shown in Table 1, we broadly divided the statements into two major areas: perceptions and attitudes towards oneself, and interpersonal relationships and environment. Subsequently, we classified these into positive, negative, and neutral categories, resulting in six distinct areas.

Subsequently, through the iterative reading of the statements, we excluded redundant, ambiguous, or irrelevant content that did not align with the study’s objectives. Throughout this process, we held multiple consultations and meetings with six Q methodology experts, resulting in 58 statements.

Face validity involves assessing the degree to which the items are appropriate based on subjective judgements from laypersons rather than experts [44]. For the face validity check, 11 middle-aged female breast cancer patients evaluated the statements on a Likert scale ranging from strongly agree (5 points) to strongly disagree (1 point). After removing 18 statements with an average score below 4.00 or a standard deviation greater than 1.00, we selected 40 Q statements. 

Content validity evaluates how well the test items represent what the study intends to measure, assessed through logical reasoning and analysis by content experts [45]. Four Q methodology experts reviewed the study for content validity, resulting in the final confirmation of 40 statements.

Finally, we conducted reliability testing of the final Q sample with five research participants, including four individuals in their 40s and one in her 50s, with two in stage 1, two in stage 2, and one in stage 3 of breast cancer. The test yielded an average correlation coefficient of r = 0.888, confirming the final selection of 40 items. Figure 2 illustrates the detailed selection process for the Q sample, while Table 2 presents the final 40 Q statements.

### 2.2. P Sample

The P sample refers to the participants engaged in Q sorting according to the research topic [46]. In the Q methodology, which adheres to the principle of small samples, various sample sizes are feasible, typically ranging from 30 to 50 participants [42,47]. We obtained approval for this study for research cooperation and participant recruitment postings by submitting the research proposal and IRB (Institutional Review Board) approval (DUIRB-202204-03) to the administrator of an online community exclusively for breast cancer patients. Through this process, we solicited applications for participation from women aged 40 to 64 with breast cancer. Due to the coronavirus 2019 (COVID-19) pandemic and the nature of the disease, we conducted the research remotely.

We selected the P sample by explaining the research objectives and methods via phone and online, followed by detailed explanations of the research objectives, content, methods, and procedures to patients who voluntarily agreed to participate. The final P sample, who consented to participate in the Q sorting study, included seventeen participants in their 40s, eighteen in their 50s, and four in their 60s. The distribution by cancer stage was as follows: twelve participants were in stage 1, fourteen in stage 2, twelve in stage 3, and one in stage 4.

### 2.3. Q Sorting

Q sorting involved 39 participants from the P sample who sorted 40 Q statements into a Q sorting grid based on their levels of agreement, derived from their subjective experiences, values, beliefs, and thoughts, following a normal distribution pattern. The Q-sorting procedure is in Figure 3.

We facilitated this process by explaining the Q statements, sorting grids, and examples to the participants online, with additional explanations provided via phone regarding the sorting methodology. After thoroughly explaining the Q classification process, participants engaged in Q sorting.

We asked participants to classify the entire set of statements into three categories—agree, neutral, and disagree—on an 11-point grid scale, where the far right represented the highest level of agreement and the far left represented the highest level of disagreement. Following this, we instructed participants to provide reasons for choosing the two statements they most agreed and disagreed with, located at each end of the sorting grid. This method is effective for identifying characteristics by type. The average sorting time per participant ranged from approximately 30 to 50 min. Figure 4 illustrates the Q distribution.

### 2.4. Data Analysis

The data obtained from the Q-sorting were subjected to principal component factor analysis using the QUANAL programme which was developed by Norman Van Tubergen in the 1960s for mainframe platform Based on the resulting Z-scores, we extracted factors using an eigenvalue threshold of 1.00 [42]. Factor weights indicate how much one’s characteristics represent the types [42]. To identify the characteristics of subjective types related to the experience of middle-aged women with breast cancer, we subsequently interviewed P samples that represented the highest factor weights in each type. The five P samples with the highest factor weights by type included one participant in their 40s, three in their 50s, and one in their 60s, with cancer stages comprising two in stage 1, two in stage 2, and one in stage 3. Subsequent interviews with P samples with high factor weights can enhance understanding and provide greater qualitative depth.

## 3. Research Results

### 3.1. Analysis of Results

The study derived five types, as shown in Table 3. The eigenvalues for the individual types were 11.86 for Type 1, 3.54 for Type 2, 3.20 for Type 3, 2.69 for Type 4, and 1.88 for Type 5. The cumulative variance was 59.43.

### 3.2. Characteristics of Perception Types

Examining the factor weights of subjective types related to the experiences of middle-aged women with breast cancer, the representative P samples with the highest factor weights for each type were as follows: P24 with a factor weight of 1.9371 for Type 1, P15 with 1.4540 for Type 2, P34 with 1.5733 for Type 3, P14 with 2.5126 for Type 4, and P12 with 2.4584 for Type 5. General characteristics and factor weights of P samples by type are in Table 4.

#### 3.2.1. Type 1: Embracing a New Life

The core characteristic of Type 1 (Table 5) is a high appreciation for life and a broadened perspective, actively pursuing life post-diagnosis compared to pre-cancer times. Statements with strong agreement include Q1, “I am grateful just to be alive right now” (Z = 1.89), and Q9, “Breast cancer has been a turning point that allowed me to live a new life” (Z = 1.72).

P24, exemplifying Type 1 characteristics, stated, “Before breast cancer, I sought meaning and satisfaction in faithfully fulfilling multiple roles, but after the diagnosis, simply being alive became the reason and satisfaction of life itself”. In-depth interviews post-Q-sorting revealed the following statements from Type 1 participants:

I lived too hard. When I got cancer, it felt like it was bound to happen, so accepting the fact was easier for me. The initial shock of the diagnosis was natural. After overcoming the initial fear, I’ve come to appreciate each day just for being alive. Additionally, I find myself grateful for life and even volunteering.(P24)

Rather than desires about death or wanting to live, I realised I’m still alive, and there are still things for me to do. I decided to live gratefully and joyfully. Changing my mindset like that, I’m not afraid of death anymore, and I dare to speak up. In doing so, I avoid harbouring resentment or greed and always try to shift my thoughts positively.(P29)

I realised that life and death are not areas I can control. So I thought it would be better to do what I can. When people ask me questions, I sincerely answer with all the knowledge I have. I hope I can be of some help, even if it’s small.(P11)

These examples indicate that Type 1 participants have positively expanded their perspectives on themselves and others compared to pre-diagnosis.

#### 3.2.2. Type 2: Finally Focusing on Myself

The core characteristic of Type 2 (see Table 6) is prioritising self-care and focusing on personal well-being post-breast cancer diagnosis. Statements with strong agreement include Q26, “Now, my life’s priority is ‘myself” (Z = 1.36), and Q21, “I encourage and support myself more than before” (Z = 1.28).

Post-Q-sorting interviews revealed Type 2 participants saying:

When I’m sick, it only upsets me. I thought it would be less of a burden on my family if I were healthy. Instead of asking my son and husband to take care of me, I think it’s better for me to manage myself and live healthily.(P15)

I used to think the company and home couldn’t run without me, but I learned they can manage fine. Since then, I’ve let go of everything and decided to take care of myself first. If I’m sick, nothing else matters.(P17)

In addition, P23 said, “I need to take care of myself. I really have to take care of my body, my health. That’s the first priority now”. These statements show a cognitive shift towards considering oneself the most important person in life and initiating self-care based on self-compassion post-diagnosis.

#### 3.2.3. Type 3: Struggling Amidst Confusion

The core characteristic of Type 3 (see Table 7) is experiencing positive and negative thoughts and emotions in nearly equal proportions during breast cancer treatment. Statements with strong agreement include Q1, “I am grateful just to be alive right now” (Z = 2.23), and Q13, “When I think about metastasis or recurrence, I feel overwhelmed with fear” (Z = 1.38).

Post-Q-sorting interviews revealed Type 3 participants stating:

No matter the type of cancer, once you become a cancer patient, living with the fear of recurrence and metastasis is a common sentiment. I try to stay positive and forget, but being a cancer patient, I often feel trapped in this anxiety, and it makes me feel unhappy.(P34)

As I get older, I just need to live as a human being. I hate it when people dismiss my concerns about being a woman. But there’s truth to it. So, since this is how things are, I just aim to live as a good human being.(P4)

In addition, P26 stated, “Sometimes I forget about being sick for a while, but when the anxiety about recurrence strikes, I feel like I’m sinking into a deep hole”. These statements indicate that participants experience significant psychological distress due to anxiety about disease progression and existing conditions while appreciating life’s dualistic emotions.

#### 3.2.4. Type 4: Sorrowfully Withdrawing from the World

The core characteristic of Type 4 (see Table 8) is sadness over the loss of femininity and sense of purpose during breast cancer surgery and treatment, withdrawing from external scrutiny. Statements with strong agreement include Q37, “It is difficult for me to go to public places like saunas or swimming pools where I have to undress” (Z = 2.02), and Q39, “Changes in my chest and body shape undermine my self-esteem” (Z = 1.29). P14, exemplifying Type 4 characteristics, stated, “I feel deeply hurt by being treated as a spectacle or a topic of discussion due to the loss of my breasts. It feels like others see me as abnormal”.

Post-Q-sorting interviews revealed Type 4 participants saying,

Even though I’m older, I’m still a woman. Because of the treatment, they removed my nipples. I took tamoxifen, which thickened my endometrial lining, so they removed my uterus too. I’ve lost everything that made me a woman—my uterus, ovaries, and breasts. There are times when I feel submerged in thoughts of whether I am still a woman. Every time I shower and see my body, it doesn’t feel good. My self-esteem has plummeted.(P14)

I underwent another surgery. I don’t like my body; how could others? On top of that, I’ve gained so much weight due to the side effects of tamoxifen. Because of the treatment, I quit my job, and I’m increasingly disliking meeting people and going out.(P20)

In addition, P19 said, “It dries up all the female hormones. I’m angry again. The surgery, too. How could they do this to my chest? What am I now?” These statements indicate significant psychological distress among participants due to the loss of femininity during treatment.

#### 3.2.5. Type 5: Pushed to the Edge of Fear

The core characteristic of Type 5 (see Table 9) is experiencing strong fear and terror of death and the associated pain during breast cancer treatment compared to before diagnosis. Statements with strong agreement include Q13, “When I think about metastasis or recurrence, I feel overwhelmed with fear” (Z = 1.87), and Q6, “I often feel fearful about not knowing when I might die” (Z = 1.20). P12, exemplifying Type 5 characteristics, stated, “I feel hurt when close people do not try to understand my internal pain and fear”.

Post-Q-sorting interviews revealed Type 5 participants saying, “If it recurs, it’s really the end. I’m scared of experiencing real pain. That’s why I asked my husband to let me euthanize myself in Switzerland. I’m afraid of dying, but I’m more afraid of dying in pain” (P12), and “We’re really dry every six months. When they suspected metastasis last month, I thought it was the end of the world” (P25). In addition, P36 stated,

Without realizing it, I become concerned over minor changes or pain in my body, worrying if they might indicate a recurrence or metastasis, becoming hypersensitive. The anxiety overwhelms me, especially when I hear about other patients experiencing recurrence or metastasis.

These statements indicate that participants experience significant fear and anxiety about physical pain and death during treatment.

Table 10 shows the consensus statements, represented by Q22, Q15, and Q20.

## 4. Discussion

This study categorised the subjective experiences of middle-aged female breast cancer patients into five types (Figure 5), ranging from post-traumatic growth to post-traumatic stress disorder (PTSD) characteristics.

Type 1’s (Embracing a New Life) characteristics include a more accepting and open-minded attitude towards oneself and the world after the illness compared to before. This view suggests that the experience of illness is a transformative crisis, leading to a more positive outlook on life rather than a pessimistic one. In other words, Type 1 respondents appear to experience post-traumatic growth as a result of the positive impact of deliberate rumination. This perspective is consistent with the findings of Park et al. [16], Park [48], Hong [37], Karlsson et al. [49], and Menger et al. [28]. Therefore, helping Type 1 patients should include counselling support and programmes based on positive psychology to sustain their currently achieved mature psychological well-being. Such support could foster internal joy and satisfaction while enhancing prosocial relationships and behaviours [50].

Type 2’s (Finally Focusing on Myself) characteristics include a shift in consciousness towards protecting and caring for oneself, which participants neglected due to social roles. This characteristic indicates that patients have recognised the need for self-compassion, rooted in self-kindness and understanding, as they work through and accept their trauma. In other words, self-compassion is crucial in promoting post-traumatic growth and represents a transitional stage towards such growth. This perspective aligns with the findings of Kang [51], Lim [52], Hur [36], Pinto–Gouveia et al. [53], and Yousefi and Masoumi [54]. Deliberate rumination based on self-compassion is important for Type 2 patients to achieve post-traumatic growth. Thus, providing psychological support and education to strengthen self-compassion is essential for fostering inner growth and post-traumatic development in these patients.

Type 3’s (Struggling Amidst Confusion) characteristics include persistent confusion due to ambivalent thoughts and positive and negative emotions experienced during the illness process. These characteristics indicate that patients struggle, repeatedly falling and rising as they contend with these conflicting feelings. In other words, Type 3 respondents appear to exhibit a blend of post-traumatic growth and post-traumatic stress disorder. This perspective aligns with the findings of Park et al. [16], Ahn and Suh [35], Yun and Song [55], and Joo and Kim [56]. Therefore, for Type 3 patients, it is crucial to support them in balancing negative emotions and positive thoughts rather than solely promoting a positive mindset. Psychological counselling interventions, such as mindfulness, could help them recognise their current situation objectively. This approach could lead to improved self-understanding and self-regulation.

Type 4’s (Sorrowfully Withdrawing from the World) characteristics include identity confusion and a loss of existential meaning due to physical damage during the illness. This characteristic indicates that patients view their appearance and existence negatively, creating feelings of alienation and withdrawal from the world. In other words, Type 4 respondents appear to experience difficulties similar to post-traumatic stress disorder, marked by significant psychological distress and pain. This perspective aligns with the findings of Lim [52], Brunet et al. [17], Fallbjörk et al. [43], Koçan and Gürsoy [14], and Lindwall and Bergbom [57]. Therefore, Type 4 patients require intensive individual counselling, such as trauma therapy and grief counselling, provided by specialised counsellors tailored to the individual’s needs, particularly those experienced in breast cancer care.

Type 5’s (Pushed to the Edge of Fear) characteristics include intense fear and distressing emotions due to anticipating death or pain, along with a resigned outlook towards others. These characteristics indicate that patients experience extreme anxiety, feel as though they are losing control, and exhibit excessive fear during their illness. In other words, Type 5 respondents appear to experience post-traumatic stress disorder, marked by persistent and intrusive anxiety and fear, as well as negative cognitions and emotions. This perspective aligns with the studies of Kazlauskiene and Bulotiene [26], Maheu et al. [58], Sharpe et al. [11], Vickberg [59], Waldrop et al. [24], and Xin et al. [60]. Therefore, Type 5 patients need specific programmes utilising cognitive-behavioural therapy to modify dysfunctional thoughts and beliefs and to promote a more positive outlook. Such interventions could reduce their anxiety and fear.

## 5. Limitations

While this study has significant implications, it also has the following limitations. First, the study focused on middle-aged women between 40 and 65. However, even within this group, there are differences in characteristics and incidence rates based on age, presenting a limitation in exploring these variations in detail. Future studies should conduct more closely targeted research by subdividing patients into specific age groups to obtain more detailed information. Second, the researchers conducted this study in Korea, and the participants did not disclose the specific regions where they reside. Therefore, future research should examine how breast cancer patients perceive their experiences across various cultures and regions.

Third, this study exclusively employed the Q methodology. While useful, this approach may lack depth in exploring patients’ experiences thoroughly. Subsequent research combining qualitative methods, such as life history studies, could provide a richer understanding of patients’ experiences. Moreover, the use of self-reported data from breast cancer patients in this study may introduce potential bias, as respondents could underreport or exaggerate their feelings due to factors like social desirability. Therefore, further studies that can improve the validity of such self-reported data may be necessary.

Fourth, this study examined patients’ subjectivity at a single time point. This boundary limits understanding the diverse trajectories of their illness experiences over time. Follow-up longitudinal studies that distinguish patients based on their disease stages could better explore psychological and emotional support needs and provide insights for improvement strategies.

Lastly, this study utilised Q methodology focused on typifying perceptions based on a small sample size. This approach presents limitations in generalising the findings. Therefore, it is necessary to scale the study’s results to a larger sample to lay the groundwork for generalisation research. Through such processes, researchers could apply the findings of this study to a more diverse population of breast cancer patients, uncovering differences according to socioeconomic status, education, or other demographic factors.

## 6. Conclusions and Recommendations

This study utilised the Q methodology to categorise the subjective experiences of middle-aged female breast cancer patients into five distinctive types: Embracing a New Life (Type 1), Finally Focusing on Myself (Type 2), Struggling Amidst Confusion (Type 3), Sorrowfully Withdrawing from the World (Type 4), and Pushed to the Edge of Fear (Type 5). Based on these findings, the study derived specific conclusions for each type.

First, for Type 1, characterised as Embracing a New Life, these patients can adopt an accepting attitude towards themselves and the world, pursuing gratitude for life and quality relationships. This type demonstrates characteristics of post-traumatic growth that transcend personal experiences. Respondents of this type benefit from deliberate rumination, a process of post-traumatic growth, accepting a changed world, and experiencing well-being and compliance through wisdom. Therefore, we suggest establishing programmes connected to counselling support based on positive psychology to maintain the psychological well-being achieved by Type 1. Positive psychology, as advocated by Seligman [50], expands internal positive psychology to foster joy and satisfaction while supporting overcoming adversity and self-crafted happiness. This support enables interpreting daily experiences as benefits and blessings, enhancing societal contributions and overall happiness. Such an approach from the perspective of positive psychology could help expand the productive aspects of life for Type 1 patients.

Second, the Finally Focusing on Myself type (Type 2) shows a significant shift in consciousness where patients prioritise understanding and protecting themselves after a prolonged struggle with breast cancer. This transformation is deeply linked with self-compassion, involving self-kindness and an understanding that fosters caring for a healthy mind. Type 2 patients begin to prioritise their health and manage their minds amidst their cancer journey, showing reduced self-criticism and growing acceptance of themselves, enhancing their connectivity with others.

For Type 2 respondents, facilitating deliberate reflection through self-compassion to enable a complete transition to post-traumatic growth is crucial. This shift involves strengthening self-compassion, which could serve as a precursor to post-traumatic growth, providing psychological support or education to help expand this growth.

Third, the “Struggling Amidst Confusion” type (Type 3) prominently displays ambivalent feelings, where positive and negative thoughts coexist. This ambivalence indicates an intermingling of post-traumatic growth and post-traumatic stress disorder. It is particularly important to note that Type 3 patients, compared to other types, generally have a shorter duration of illness. Those categorised under Type 3 are relatively recently diagnosed, which could amplify the shock and confusion experienced. Managing and regulating these relations requires helping patients maintain positivity—even if it must be encouraged—and providing opportunities to embrace and objectively view their current emotions and thoughts rather than rushing to overcome their grief and anxiety.

For example, approaches like mindfulness, which promotes decentred thinking, can help Type 3 patients observe their condition and thoughts from a distance. This psychological and counselling approach, based on mindfulness, allows patients to take an observer’s perspective of their experiences of breast cancer, including thoughts, emotions, actions, and motivations. They can achieve deep self-understanding, acceptance, and self-regulation by stepping back and contemplating their situation.

Fourth, the “Sorrowfully Withdrawing from the World” type (Type 4) is distinctly visible in middle-aged female breast cancer patients who, due to the physical damage incurred during the treatment process, feel alienated from the world, leading to withdrawal from social relationships. This alienation stems from sadness, negative thinking, and a rejectionist attitude. Type 4 respondents are hypersensitive to how they and their altered bodies are perceived, resulting in a view of themselves as incomplete and insecure, and thus, they experience significant psychological pressure and pain similar to that observed in post-traumatic stress disorder.

Therefore, it is essential to approach these patients with dense psycho-emotional interventions, such as trauma therapy and bereavement counselling, tailored to the level of their scars to mediate their sense of loss and assist in reestablishing their identities. This process requires the connection with specialised counsellors for breast cancer patients, suggesting the need for individualised counselling plans and personal counselling sessions for each patient. An individualised approach, helping in the reestablishment of their identity, can provide these patients with control over renegotiating and repositioning the image of their trauma-affected bodies. This acceptance of their surgery-altered bodies can facilitate the establishment of a new identity and enhance their quality of life.

Fifth, the “Pushed to the Edge of Fear” type (Type 5) exhibits the most intense fears within the study’s categories. These individuals experience excessive anxiety due to their anticipation of pain associated with death or the process leading to death. The persistent and pervasive anxiety and fear, along with negative cognitions and emotions exhibited by Type 5 respondents, align with post-traumatic stress disorder. The interventions for middle-aged female breast cancer patients who exhibit such high levels of negative emotion should involve psychological and counselling approaches based on cognitive restructuring. These patients often ignore positive information about their trauma from breast cancer surgery and treatment, overinterpreting negative thoughts, which leads to excessive fear and anxiety. The fears and anxieties of Type 5 respondents are rooted in dysfunctional beliefs, and finding realistic and adaptive alternatives to replace maladaptive thinking through regular and sustained anxiety intervention programmes can be highly beneficial.

Cognitive therapy-based psychological and counselling approaches for Type 5 patients aim to correct their distorted cognitions and shift towards a positive thinking system, thereby fostering a positive attitude towards life and potentially serving as a catalyst for post-traumatic growth. Various theories and methods, such as the third wave of cognitive behavioural therapy (CBT), including acceptance and commitment therapy (ACT), dialectical behaviour therapy (DBT), and mindfulness-based cognitive therapy (MBCT), can be effective in programmes that mitigate anxiety and fear among breast cancer patients [61,62,63]. Such psychological and counselling interventions, focusing on cognitive distortions caused by trauma, help alleviate fears and provide opportunities to explore life directions and values, guiding patients towards more goal-oriented and fulfilling lives.

In summary, the study offers the following recommendations. First, specialised, individual counselling is necessary, tailored to the personal characteristics of middle-aged female breast cancer patients. This study classified their subjective experiences of battling cancer into five types. This differentiation shows that these patients, facing breast cancer at a pivotal point in life, do not share the same psychological state. Particularly, the five types identified range from stages of post-traumatic growth to post-traumatic stress disorder, indicating that individuals at various stages of psychological and emotional states undergo different experiences of the disease. The findings suggest that understanding the types of middle-aged female breast cancer patients is crucial to provide the appropriate psychological and counselling interventions. Moreover, generic psychological and counselling approaches may not adequately address their needs because each patient’s perception and experience of trauma related to breast cancer vary, necessitating precise and detailed approaches. Specialised, personalised counselling and psychological approaches are essential beyond what general institutions typically offer.

Second, we suggest group counselling to enhance self-awareness and the recognition of one’s whole self among middle-aged female breast cancer patients. Although classified into five types, the diverse subjectivities and differentiated experiences identified in the study share commonalities in diagnosis and treatment processes, suggesting that shared experiences exist. Group counselling could facilitate rich, dynamic interactions and understanding, helping patients expand their perspectives through decentred thinking. This approach would enhance patients’ psychological well-being, enabling them to find psychological stability and learn from the attitudes of others living positively, thereby improving their quality of life.

Third, the study’s findings confirm that the experiences of middle-aged female breast cancer patients coexist with positive and negative experiences, similar to conclusions drawn in previous research by Kang [51], Lim [64], and Hur [36], which approached the topic from nursing and social welfare perspectives. The depth of subjectivity in psychological and emotional aspects particularly points to a significant finding: despite the clear physical and psychological trauma from the physical pain, loss, challenging treatments, and long duration of the illness, these women also have distinctly positive perspectives on new aspects of life. These findings underscore the importance of reaffirming patients’ experiences, highlighting the need for discussions on psychological and emotional approaches that enhance positivity and reduce negativity for breast cancer patients. They also call for further multidisciplinary research to continue exploring these dimensions.

## Figures and Tables

**Figure 1 healthcare-12-01873-f001:**
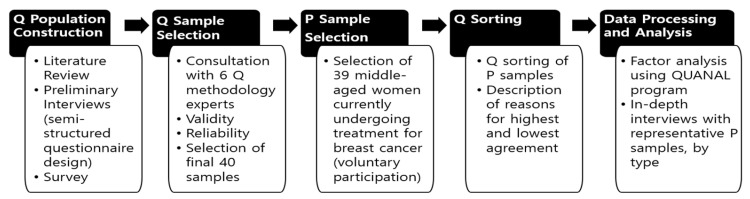
Research procedure.

**Figure 2 healthcare-12-01873-f002:**
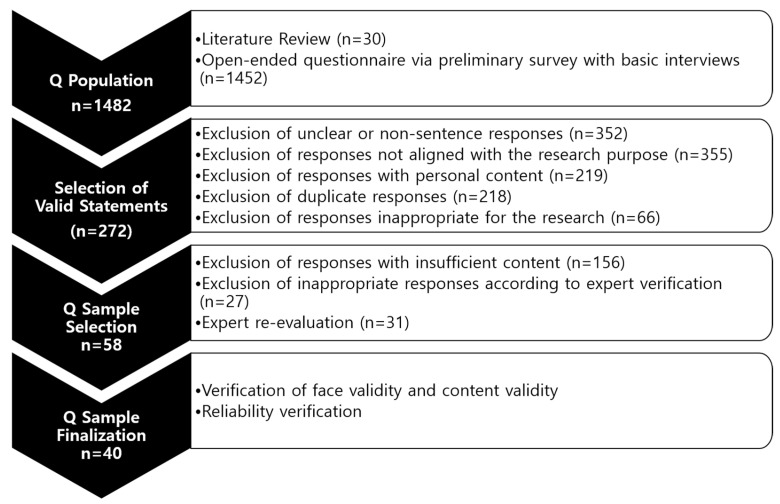
Q sample selection process.

**Figure 3 healthcare-12-01873-f003:**
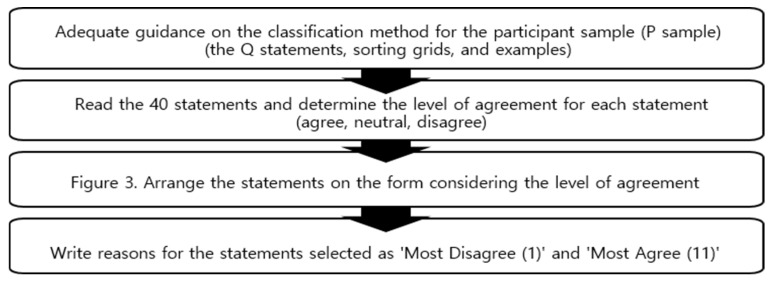
Q sorting process.

**Figure 4 healthcare-12-01873-f004:**
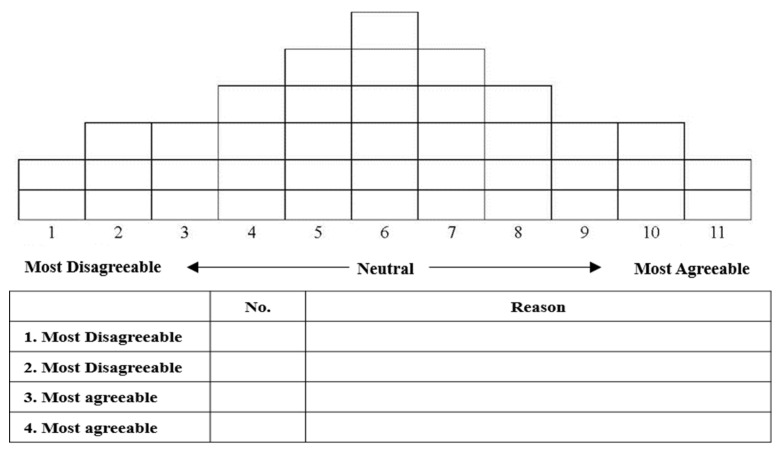
Q sorting grid and Q sorting sheet with reasons.

**Figure 5 healthcare-12-01873-f005:**
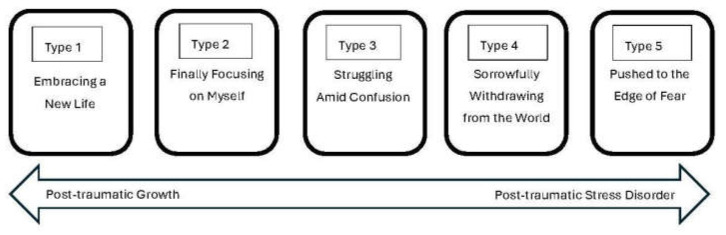
A spectrum of perception types.

**Table 1 healthcare-12-01873-t001:** Statement categories.

	Perceptions and Attitudes Towards Oneself	Perception and Attitude Towards Relationships and Environment
Positive	Discovering small happiness Gratitude for being alive Turning point in life	Value of family Empathy for fellow patients Psychological resilience
Negative	Fear of recurrence and death Shame about bodily changes Loss of hope for an uncertain future	Resentment towards family Unsettled feeling of being misunderstood Loneliness due to isolation
Neutral	Facing reality Stoicism	Acknowledgement of others’ lives Respectful living

**Table 2 healthcare-12-01873-t002:** Q sample.

N	Q Statements
Q1	I am grateful just to be alive right now.
Q2	I feel fortunate that breast cancer, with its higher survival rates compared to other cancers, happened to me.
Q3	I have come to appreciate the small joys of daily life that I didn’t notice before.
Q4	The treatment process for breast cancer is too painful.
Q5	I feel sad because my physical strength has declined so much that my daily life feels shattered.
Q6	I often feel fearful about not knowing when I might die.
Q7	I try to calm myself by thinking, “This too shall pass”.
Q8	When I look back on my life, I try to accept that cancer was something I was meant to have.
Q9	Breast cancer has been a turning point that allowed me to live a new life.
Q10	I am more proactive in expressing my thoughts and emotions than before.
Q11	I have developed a stronger attachment to life than before I got sick.
Q12	To distract myself from forgetting the reality of being a breast cancer patient, I find things on which to focus.
Q13	When I think about metastasis or recurrence, I feel overwhelmed with fear.
Q14	Even minor pain or physical changes make me worry excessively about my health.
Q15	Due to an uncertain future, I have abandoned my life goals.
Q16	If breast cancer is my fate, I think I should accept it.
Q17	I strive to let go of things and desires to which I used to cling.
Q18	I have started reflecting on my life instead of just charging forward.
Q19	Breast cancer has allowed me to find relaxation in my otherwise busy life.
Q20	Thinking I may not have much time left, I do my best in my tasks.
Q21	I encourage and support myself more than before.
Q22	I want to focus more on the present moment rather than the future.
Q23	Feeling that I have to handle and overcome cancer alone makes me feel lonely.
Q24	I pity myself for being sick and anxious about breast cancer.
Q25	Despite feeling relieved to have gotten this far, I cannot help feeling anxious.
Q26	Now, my life’s priority is “myself”.
Q27	Believing that health is most important, I am trying to change my diet and exercise habits.
Q28	I am grateful to my family (husband, children, parents, siblings, etc.) who support me.
Q29	Since being diagnosed with breast cancer, I have grown closer to my family.
Q30	I worry about how my family will live without me.
Q31	I want to witness my family grow and grow old together.
Q32	Even my family doesn’t fully understand my suffering.
Q33	I have been hurt by acquaintances who tried to comfort me.
Q34	I no longer maintain relationships with friends I feel do not understand me.
Q35	I feel discouraged by people around me who treat me as a patient.
Q36	Now, the most important people who provide me with psychological support, stability, information, etc., are fellow breast cancer patients.
Q37	It is difficult for me to go to public places like saunas or swimming pools where I must undress.
Q38	I find listening to people telling me to think positively is much harder.
Q39	Changes in my chest and body shape undermine my self-esteem.
Q40	I dislike the idea of people around me knowing that I am a breast cancer patient.

**Table 3 healthcare-12-01873-t003:** Eigenvalues and explanatory variances in the classification of five types.

Content/Type	Type 1	Type 2	Type 3	Type 4	Type 5
Eigenvalues	11.86	3.54	3.20	2.69	1.88
Variance (%)	30.42%	9.08%	8.19%	6.90%	4.83%
Cumulative (%)	30.42%	39.50%	47.70%	54.59%	59.43%

**Table 4 healthcare-12-01873-t004:** Persons and factor weights by type.

Type	P #	Age	Stage	Duration of Treatment	Cancer Metastasis	Factor Weight
Type 1 (N = 13)	1	48	2	7 years	O	0.7127
	2	52	3	1 year 1 month	O	1.1638
	5	42	3	3 years 1 month	O	1.6369
	6	43	3	9 months	O	0.7766
	7	55	4	3 years 10 month	O	1.2371
	9	54	1	2 years 6 months	X	0.7345
	11	50	3	6 years	X	0.8619
	13	54	2	1 year 6 months	X	0.4932
	22	48	1	1 year 5 months	X	1.1143
	24	54	3	9 months	X	1.9371
	27	52	3	6 months	X	0.6091
	29	53	1	3 years	X	1.6828
	35	61	2	2 years 10 months	X	1.1930
Type 2 (N = 6)	3	43	1	3 years 6 months	X	0.7648
	15	55	2	4 years 8 months	O	1.4540
	17	40	3	1 year 5 months	O	0.6942
	23	52	3	1 year	O	1.1938
	31	51	1	7 months	X	0.7761
	37	45	1	2 months	X	0.7311
Type 3 (N = 10)	4	56	1	6 months	X	0.2439
	16	47	2	6 months	O	1.2001
	18	59	3	7 years 1 month	O	0.5215
	21	49	2	1 year 2 months	O	0.9271
	26	47	3	9 months	X	1.3674
	30	60	1	2 years 8 months	X	0.9474
	32	52	1	5 years 10 months	X	0.9709
	33	54	2	1 year 6 months	X	1.0602
	34	43	2	1 year 2 months	O	1.5733
	38	46	2	4 months	O	1.1003
Type 4 (N = 5)	10	49	2	2 years	X	0.5661
	14	64	1	2 years 9 months	X	2.5126
	19	64	1	4 months	X	0.9028
	20	41	2	1 year 1 month	X	1.0662
	39	46	3	3 years 9 months	O	0.9047
Type 5 (N = 5)	8	47	2	2 years 6 months	X	0.6623
	12	52	1	3 years 6 months	X	2.4584
	25	51	2	1 year 9 months	X	1.1328
	28	53	3	8 months	O	0.5852
	36	48	2	2 years 8 months	O	0.9701

Note: X = No Cancer Metastasis, O = Cancer Metastasis Present.

**Table 5 healthcare-12-01873-t005:** The statements of Type 1 and Z scores (≥1.00).

No.	Statement	Z Score
1	I am grateful just to be alive right now.	1.89
9	Breast cancer has been a turning point that allowed me to live a new life.	1.72
19	Breast cancer has allowed me to find relaxation in my otherwise busy life.	1.45
3	I have come to appreciate the small joys of daily life that I didn’t notice before.	1.38
22	I want to focus more on the present moment rather than the future.	1.25
35	I feel discouraged by people around me who treat me as a patient.	−1.02
23	Feeling that I have to handle and overcome cancer alone makes me feel lonely.	−1.13
6	I often feel fearful about not knowing when I might die.	−1.44
24	I pity myself for being sick and anxious about breast cancer.	−1.72
15	Due to an uncertain future, I have abandoned my life goals.	−1.76

**Table 6 healthcare-12-01873-t006:** The statements of Type 2 and Z score (≥1.00).

No.	Statement	Z Score
22	I want to focus more on the present moment rather than the future.	1.63
26	Now, my life’s priority is “myself”.	1.36
13	When I think about metastasis or recurrence, I feel overwhelmed with fear.	1.33
21	I encourage and support myself more than before.	1.28
9	Breast cancer has been a turning point that allowed me to live a new life.	1.12
35	I feel discouraged by people around me who treat me as a patient.	−1.03
24	I pity myself for being sick and anxious about breast cancer.	−1.42
39	Changes in my chest and body shape undermine my self-esteem.	−1.62
15	Due to an uncertain future, I have abandoned my life goals.	−1.73
8	When I look back on my life, I try to accept that cancer was something I was meant to have.	−2.09

**Table 7 healthcare-12-01873-t007:** The statements of Type 3 and Z scores (≥1.00).

No.	Statement	Z Score
1	I am grateful just to be alive right now.	2.23
13	When I think about metastasis or recurrence, I feel overwhelmed with fear.	1.38
31	I want to witness my family grow and grow old together.	1.35
7	I try to calm myself by thinking, “This too shall pass”.	1.32
14	Even minor pain or physical changes make me worry excessively about my health.	1.20
33	I have been hurt by acquaintances who tried to comfort me.	−1.06
40	I dislike the idea of people around me knowing that I am a breast cancer patient	−1.38
15	Due to an uncertain future, I have abandoned my life goals.	−1.49
35	I feel discouraged by people around me who treat me as a patient.	−1.68
39	Changes in my chest and body shape undermine my self-esteem.	−1.76

**Table 8 healthcare-12-01873-t008:** The statements of Type 4 and Z scores (≥1.00).

No.	Statement	Z Score
37	It is difficult for me to go to public places like saunas or swimming pools where I must undress.	2.02
25	Despite feeling relieved to have gotten this far, I cannot help feeling anxious.	1.90
13	When I think about metastasis or recurrence, I feel overwhelmed with fear.	1.49
39	Changes in my chest and body shape undermine my self-esteem.	1.29
33	I have been hurt by acquaintances who tried to comfort me.	1.02
21	I encourage and support myself more than before.	−1.02
11	I have developed a stronger attachment to life than before I got sick.	−1.26
15	Due to an uncertain future, I have abandoned my life goals.	−1.77
20	Thinking I may not have much time left, I do my best in my tasks.	−1.96
6	I often feel fearful about not knowing when I might die.	−2.13

**Table 9 healthcare-12-01873-t009:** The statements of Type 5 and Z scores (≥1.00).

No.	Statement	Z Score
13	When I think about metastasis or recurrence, I feel overwhelmed with fear.	1.87
32	Even my family doesn’t fully understand my suffering.	1.34
6	I often feel fearful about not knowing when I might die.	1.20
27	Believing that health is most important, I am trying to change my diet and exercise habits.	1.11
14	Even minor pain or physical changes make me worry excessively about my health.	1.11
39	Changes in my chest and body shape undermine my self-esteem.	−1.14
28	I am grateful to my family (husband, children, parents, siblings, etc.) who support me.	−1.19
29	Since being diagnosed with breast cancer, I have grown closer to my family.	−1.34
30	I worry about how my family will live without me.	−1.60
5	I feel sad because my physical strength has declined so much that my daily life feels shattered.	−2.10

**Table 10 healthcare-12-01873-t010:** Consensus statements of individual types.

No.	Statement	Z Score
22	I want to focus more on the present moment rather than the future.	1.074
15	Due to an uncertain future, I have abandoned my life goals.	−1.610
20	Thinking I may not have much time left, I do my best in my tasks.	−1.176

## Data Availability

The datasets generated and/or analysed during the current study are available from the corresponding author upon reasonable request.

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
