# Peer review of "Categorising Subjective Perceptions of Middle-Aged Breast Cancer Patients Using Q Methodology"

_healthcare, 2024, doi:10.3390/healthcare12181873_

Round 1
Reviewer 1 Report
Comments and Suggestions for Authors
The authors have provided valuable insights into the psychological profiles of middle-aged female breast cancer patients by identifying and categorizing five distinct perception types using Q methodology. Their analysis offers a detailed understanding of how these patients experience their illness, ranging from embracing new life opportunities to feeling overwhelmed by fear and despair. While the study provides valuable insights into the psychological profiles of middle-aged female breast cancer patients, there are several potential drawbacks to consider.
1. The limited sample size may restrict the generalizability of the findings. A larger sample could yield more robust results and reveal additional viewpoint types, which the authors acknowledge as a limitation.
2. Drawbacks from the Q methodology as the study used a fixed set of statements, which might have limited how the participants could respond.
3. The study may be focused on a particular cultural or demographic group, limiting the applicability of the results to other populations.
4. Although the study proposes psychological counseling interventions tailored to each observation type, it lacks detailed strategies for implementing those in clinical practice and does not evaluate their effectiveness.
5. The use of self-reported data may introduce potential bias, as patients might underreport or overstate their feelings due to factors such as social desirability.
6. If the sample is homogeneous in terms of socio-economic status, education, or other demographic factors, the findings may not be applicable to a more diverse population of breast cancer patients.
7. Authors should rewrite the conclusion part more clear and elaborate on the applications of this study to the future research
Author Response
REVIEWER 1
Comments and Suggestions for Authors
The authors have provided valuable insights into the psychological profiles of middle-aged female breast cancer patients by identifying and categorizing five distinct perception types using Q methodology. Their analysis offers a detailed understanding of how these patients experience their illness, ranging from embracing new life opportunities to feeling overwhelmed by fear and despair. While the study provides valuable insights into the psychological profiles of middle-aged female breast cancer patients, there are several potential drawbacks to consider.
- The limited sample size may restrict the generalizability of the findings. A larger sample could yield more robust results and reveal additional viewpoint types, which the authors acknowledge as a limitation.
Response: In response to the reviewer’s suggestion, we revised the limitations section of our manuscript as follows (lines 522–528):
Lastly, this study utilised Q methodology focused on typifying perceptions based on small sample size. This approach presents limitations in generalising the findings. Therefore, it is necessary to scale the study’s results to a larger sample to lay the groundwork for generalisation research. Through such processes, researchers could apply the findings of this study to a more diverse population of breast cancer patients, uncovering differences according to socioeconomic status, education, or other demographic factors.
We hope this revision adequately addresses the reviewer’s concerns.
- Drawbacks from the Q methodology as the study used a fixed set of statements, which might have limited how the participants could respond.
Response: Thank you for your valuable feedback. We detailed the validation process in the methodology section to address the limitations mentioned. Additionally, we conducted in-depth interviews with participants with high factor loadings after the Q-sorting process to understand the reasons behind their agreement or disagreement with specific statements. Although we previously presented these interviews, we now supplemented the research findings section with more detailed descriptions to enhance the clarity and depth of our analysis.
We hope these revisions meet the reviewer’s expectations.
- The study may be focused on a particular cultural or demographic group, limiting the applicability of the results to other populations.
Response: Thank you for your feedback. We acknowledge conducting this study in South Korea and could not specify the exact regions where the participants reside. In response to the reviewer’s comments, we revised the limitations section to highlight the need for future research to explore how breast cancer patients perceive their experiences across different cultures and regions. This approach will provide a broader understanding of the factors influencing their perceptions.
We hope these revisions adequately address the reviewer’s concerns.
- Although the study proposes psychological counseling interventions tailored to each observation type, it lacks detailed strategies for implementing those in clinical practice and does not evaluate their effectiveness.
Response: To address the reviewer’s comments, we revised and expanded the “Conclusion” section, renaming it to “Conclusion and Recommendations” (lines 381–502). We made significant modifications to this part of the manuscript. Please refer to the updated “Conclusion and Recommendations” section for further details. Thank you for your valuable feedback.
- The use of self-reported data may introduce potential bias, as patients might underreport or overstate their feelings due to factors such as social desirability.
Response: Paulhus (1991) addressed these concerns in his paper “Measurement and Control of Response Bias,” where he proposed various methods to enhance the reliability of self-report data. Based on these principles, we implemented several measures during the design phase of our study.
First, we ensured strict anonymity for participants, allowing them to report their emotions honestly, thereby minimizing biases due to social desirability. Additionally, we tried to enhance the reliability and validity of the self-report data. We sought face validity by consulting six experts and involving 11 middle-aged female breast cancer patients in selecting Q-statements. Four experts in Q methodology verified the content validity. We conducted inter-rater reliability tests to ensure reliability, achieving an average reliability coefficient of r = .888. Through these methodological approaches, we aimed to minimize potential biases in the self-report data and enhance the reliability and validity of our research findings.
Furthermore, during the revision process, we conducted interviews with participants with high factor loadings in each typology after the Q-sorting, and we supplemented the manuscript with these findings.
- If the sample is homogeneous in terms of socio-economic status, education, or other demographic factors, the findings may not be applicable to a more diverse population of breast cancer patients.
Response: Thank you for your insightful comments. We acknowledged our research methodology’s potential limitations and addressed these in the Limitations section. Additionally, we outlined suggestions for future research to overcome these limitations.
- Authors should rewrite the conclusion part more clear and elaborate on the applications of this study to the future research
Response: Thank you for your feedback. We revised the conclusion section accordingly in response to the reviewer’s comments.
Reviewer 2 Report
Comments and Suggestions for Authors
Title: The title effectively summarizes the study’s focus on the subjective perceptions of middle-aged breast cancer patients. However, it could be more specific by including the methodology used.
Abstract: The abstract provides a clear overview of the study's aims, methodology, key findings, and implications. It concisely summarizes the categorization of subjective perceptions and the psychological interventions proposed.
Introduction: The introduction is well-structured and provides a comprehensive background on breast cancer's physical and psychological impacts. It effectively sets the stage for the study by highlighting the need to understand the subjective experiences of middle-aged breast cancer patients.
Research Method: The methodology section is detailed and clearly outlines the use of Q methodology to categorize subjective experiences. The process of constructing the Q population, selecting Q samples, and performing Q sorting is well-explained. The inclusion of literature reviews, preliminary interviews, and subjective surveys ensures a robust approach to gathering data.
Data Analysis: The data analysis section is thorough, describing the use of principal component factor analysis and the extraction of factors based on Z-scores. The explanation of factor weights and the subsequent interviews with P samples enhances the understanding of the subjective types identified.
Results: The results section presents a detailed analysis of the five perception types identified: Embracing a New Life, Finally Focusing on Myself, Struggling Amidst Confusion, Sorrowfully Withdrawing from the World, and Pushed to the Edge of Fear. The use of Z-scores to highlight key statements for each type is effective. The inclusion of participant quotes provides valuable qualitative insights.
Discussion: The discussion is comprehensive and effectively interprets the findings in the context of existing literature. It highlights the implications for psychological and counseling interventions tailored to each perception type. The discussion also acknowledges the potential for both post-traumatic growth and post-traumatic stress disorder among breast cancer patients.
Conclusion: The conclusion succinctly summarizes the study's contributions and emphasizes the need for tailored interventions. It calls for further research to address the limitations identified.
Limitations: The limitations section is candid and identifies key areas for improvement, such as the need for more detailed age-specific research, the inclusion of qualitative methods, and longitudinal studies.
References: The references are comprehensive and up-to-date, covering relevant studies in the field of breast cancer research.
Overall Assessment: The study is well-conducted and contributes valuable insights into the subjective experiences of middle-aged breast cancer patients. The use of Q methodology is appropriate and well-executed. The article is well-written, and the findings are clearly presented and discussed.
Recommendations:
Title Improvement: Consider refining the title to reflect the methodology, e.g., “Categorizing Subjective Perceptions of Middle-Aged Breast Cancer Patients Using Q Methodology.”
Methodology Detail: While the methodology is well-detailed, a visual representation (flowchart) of the Q sorting process could enhance understanding.
Qualitative Depth: Incorporate more direct quotes from participants in the results section to provide deeper qualitative insights.
Future Research: Emphasize the need for future research to explore the identified perception types in different cultural contexts and among different age groups.
This article is a significant contribution to the field and provides a solid foundation for further research and intervention development.
Author Response
REVIEWER 2
Comments and Suggestions for Authors
Title: The title effectively summarizes the study’s focus on the subjective perceptions of middle-aged breast cancer patients. However, it could be more specific by including the methodology used.
Abstract: The abstract provides a clear overview of the study’s aims, methodology, key findings, and implications. It concisely summarizes the categorization of subjective perceptions and the psychological interventions proposed.
Introduction: The introduction is well-structured and provides a comprehensive background on breast cancer’s physical and psychological impacts. It effectively sets the stage for the study by highlighting the need to understand the subjective experiences of middle-aged breast cancer patients.
Research Method: The methodology section is detailed and clearly outlines the use of Q methodology to categorize subjective experiences. The process of constructing the Q population, selecting Q samples, and performing Q sorting is well-explained. The inclusion of literature reviews, preliminary interviews, and subjective surveys ensures a robust approach to gathering data.
Data Analysis: The data analysis section is thorough, describing the use of principal component factor analysis and the extraction of factors based on Z-scores. The explanation of factor weights and the subsequent interviews with P samples enhances the understanding of the subjective types identified.
Results: The results section presents a detailed analysis of the five perception types identified: Embracing a New Life, Finally Focusing on Myself, Struggling Amidst Confusion, Sorrowfully Withdrawing from the World, and Pushed to the Edge of Fear. The use of Z-scores to highlight key statements for each type is effective. The inclusion of participant quotes provides valuable qualitative insights.
Discussion: The discussion is comprehensive and effectively interprets the findings in the context of existing literature. It highlights the implications for psychological and counseling interventions tailored to each perception type. The discussion also acknowledges the potential for both post-traumatic growth and post-traumatic stress disorder among breast cancer patients.
Conclusion: The conclusion succinctly summarizes the study’s contributions and emphasizes the need for tailored interventions. It calls for further research to address the limitations identified.
Limitations: The limitations section is candid and identifies key areas for improvement, such as the need for more detailed age-specific research, the inclusion of qualitative methods, and longitudinal studies.
References: The references are comprehensive and up-to-date, covering relevant studies in the field of breast cancer research.
Overall Assessment: The study is well-conducted and contributes valuable insights into the subjective experiences of middle-aged breast cancer patients. The use of Q methodology is appropriate and well-executed. The article is well-written, and the findings are clearly presented and discussed.
Recommendations:
Title Improvement: Consider refining the title to reflect the methodology, e.g., “Categorizing Subjective Perceptions of Middle-Aged Breast Cancer Patients Using Q Methodology.”
Response: We made the revisions following the reviewer’s comments. Thank you.
Methodology Detail: While the methodology is well-detailed, a visual representation (flowchart) of the Q sorting process could enhance understanding.
Response: We made the revisions following the reviewer’s comments. Thank you.
Qualitative Depth: Incorporate more direct quotes from participants in the results section to provide deeper qualitative insights.
Response: We sincerely appreciate the reviewer’s insightful comments and have carefully made the revisions accordingly. Thank you for your valuable feedback.
Future Research: Emphasize the need for future research to explore the identified perception types in different cultural contexts and among different age groups.
Response: We significantly revised the manuscript in response to the reviewer’s comments.
This article is a significant contribution to the field and provides a solid foundation for further research and intervention development.
Reviewer 3 Report
Comments and Suggestions for Authors
Dear authors,
I enjoyed reading your manuscript, and I believe it provides important insights and recommendations for approaching women who have endured (and continue to endure) breast cancer.
The manuscript provides a clear, logical, well-written text.
There are some revisions that the authors need to complete:
- Consider including women as a keyword.
- A part of the results needs to be included - specifically, on page 8, under section 3.2.2 Type 2: Finally Focusing on Myself, there is no description of the result as with other sections. This absence results in Table 6 not being presented in the manuscript. You have Table 5 on page 8 and Table 7 on page 9. I recommend completing the manuscript with the missing section - 3.2.2 Type 2: Finally Focusing on Myself.
- The section 'Limitation' would benefit from changing to plural - 'Limitations'. This is because the section discusses multiple limitations, and using the plural form will accurately reflect this.
Please address the issues presented.
Regards
Author Response
REVIEWER 3
Dear authors,
I enjoyed reading your manuscript, and I believe it provides important insights and recommendations for approaching women who have endured (and continue to endure) breast cancer.
The manuscript provides a clear, logical, well-written text.
There are some revisions that the authors need to complete:
Consider including women as a keyword.
Response: We added “women” as a keyword per the reviewer’s suggestion. Thank you.
A part of the results needs to be included - specifically, on page 8, under section 3.2.2 Type 2: Finally Focusing on Myself, there is no description of the result as with other sections. This absence results in Table 6 not being presented in the manuscript. You have Table 5 on page 8 and Table 7 on page 9. I recommend completing the manuscript with the missing section - 3.2.2 Type 2: Finally Focusing on Myself.
Response: Thank you for your comments. We confirmed that the content regarding Type 2 is already in the manuscript. During this process, we provided more detailed and comprehensive citations of the study participants.
The section ‘Limitation’ would benefit from changing to plural - ‘Limitations’. This is because the section discusses multiple limitations, and using the plural form will accurately reflect this.
Response: We have revised accordingly. Thank you.